# Chromosome biorientation produces hundreds of piconewtons at a metazoan kinetochore

Anna A. Ye[1,2], Stuart Cane[1,2] & Thomas J. Maresca[1,2]

High-fidelity transmission of the genome through cell division requires that all sister kinetochores bind to dynamic microtubules (MTs) from opposite spindle poles. The application of opposing forces to this bioriented configuration produces tension that stabilizes kinetochore–microtubule (kt–MT) attachments. Defining the magnitude of force that is applied to kinetochores is central to understanding the mechano-molecular underpinnings of chromosome segregation; however, existing kinetochore force measurements span orders of magnitude. Here we measure kinetochore forces by engineering two calibrated force sensors into the *Drosophila* kinetochore protein centromere protein (CENP)-C. Measurements of both reporters indicate that they are, on average, under ∼1–2 piconewtons (pNs) of force at metaphase. Based on estimates of the number of CENP-C molecules and MTs per *Drosophila* kinetochore and envisioning kinetochore linkages arranged such that they distribute forces across them, we propose that kinetochore fibres (k-fibres) exert hundreds of pNs of poleward-directed force to bioriented kinetochores.

[1] Biology Department, University of Massachusetts, Amherst, Massachusetts 01003, USA. [2] Molecular and Cellular Biology Graduate Program, University of Massachusetts, Amherst, Massachusetts 01003, USA. Correspondence and requests for materials should be addressed to T.J.M. (email: tmaresca@bio.umass.edu).

The forces that act on the kinetochore (kt) and how they are transduced depend on the nature of kt–microtubule (MT) interactions and the identity and number of force producers. Early in mitosis, kinetochores predominantly associate with the sides of MTs. The forces produced by kinetochore-associated motors laterally interacting with MTs are transmitted into sliding chromosomes directionally within the spindle. The major kinetochore-associated force producers at this stage are the plus-end-directed centromere protein (CENP)-E and the minus-end-directed dynein each of which are capable of producing forces in the low pN range ($\sim$1–8 pNs)[1–4]. While motor-mediated forces produced during lateral kt–MT interactions facilitate proper chromosome congression[5,6], they are neither sufficient to satisfy the spindle assembly checkpoint nor to support accurate chromosome segregation during mitosis[7]. Rather, these outcomes require the formation of stable end-on kt–MT attachments that are mediated by a conserved MT-binding complex in the outer kinetochore called the KMN (KNL-1, Mis12 complex, Ndc80 complex) network[8].

End-on attached kinetochores must be able to harness the forces produced by MT dynamics to move chromosomes. MT polymerization exerts forces ($\sim$3–4 pN) similar to those produced by kinetochore motors[9], while depolymerization of a single MT has been estimated to produce forces up to 65 pN (ref. 10), an order of magnitude higher than individual motors. Poleward pulling forces applied to end-on attached kinetochores contribute to prometaphase congression, metaphase oscillations and anaphase A movements[11–15]. When biorientation is established, opposing poleward forces produce tension across sister kinetochores that stabilizes kt–MT attachments and contributes to spindle assembly checkpoint satisfaction[16–18]. There is a surprising lack of consensus about the magnitude of force that is applied to the kinetochore despite the fact that it is one of the most important force-transducing structures in the cell.

In principle, very low forces are sufficient to move non-bioriented chromosomes if the only opposing force is the viscous drag of the cytoplasm[19]. Indeed, analyses of anaphase chromosome movements in insect meiotic and mitotic cells yielded force estimates below 1 pN (refs 19,20). However, the forces required for moving mono-oriented chromosomes poleward in prometaphase amphibian cells were estimated to be considerably higher and ranged between $\sim$10 and 75 pN (ref. 21). The

discrepancy is likely a result of polar ejection forces and steric hindrance from astral MTs opposing prometaphase but not anaphase chromosome movements. Prometaphase poleward movements are mainly driven by lateral sliding along MTs by kinetochore-associated dynein[22], but differences in prometaphase (10–75 pN) and anaphase ($<$1 pN) forces cannot simply be attributed to dynein producing higher forces than is generated by depolymerizing kt–MTs during anaphase. To the contrary, a landmark study employing calibrated microneedles concluded that anaphase k-fibres in insect spermatocytes produced up to 700 pN (50 pN per MT) of poleward directed force[23]. It is worth noting that 50 pN per MT may be an overestimate since it was based on the assumption that only microtubules spanning the kinetochore and pole, estimated to be $\sim$15 out of $\sim$45 kt–MTs, could effectively produce force. It has subsequently been shown that kt–MTs do not need to be linked directly to the pole to exert forces on the kinetochore[24–26]. Nonetheless, for simplicity and consistency's sake, the original suite of per MT force estimates[27] will be referred to from here onward when discussing Nicklas' spermatocyte studies.

While the 700 pN stall force was measured in anaphase, this number may better reflect the magnitude of force applied to metaphase kinetochores since the application of opposing force with a microneedle more closely resembles the bioriented configuration than the typical anaphase scenario. However, measuring forces at bioriented kinetochores has presented a significant challenge. Researchers recently estimated that the mean pericentromeric tension at bioriented yeast kinetochores is 4–6 pN per MT (and per kinetochore since the budding yeast kinetochore binds one MT)[28]. This value fits reasonably well with in vitro optical trapping studies using purified budding yeast kinetochore proteins and isolated kinetochore particles that have reconstituted associations with single MT plus ends over forces ranging from $\sim$2 to 9 pN (reviewed in Yusko and Asbury[29]). Although forces above 20 pN have been theoretically inferred[30] and directly measured for kinetochore proteins attached to beads via extended linkers[31]. While a general consensus (within an order of magnitude) may be emerging for the forces that are applied to budding yeast kinetochores, experimental measurements in metazoans have diverged markedly. In contrast to earlier measurements of 700 pN per kinetochore (50 pN per MT)[23], a more recent optical trapping study in meiotic insect cells and

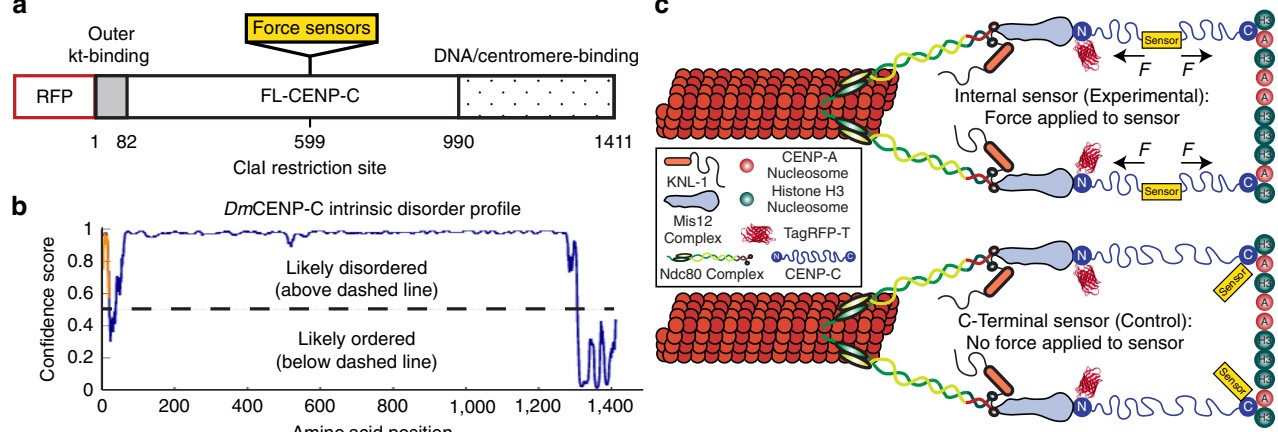

**Figure 1 | Construction of the CENP-C-based force sensors and the experimental design for force measurements. (a)** Drosophila CENP-C organization highlighting the N-terminal outer kinetochore (kt) binding domain (grey), the C-terminal DNA/centromere binding domain (polka dotted), and the placement of the internal force sensor. **(b)** Disorder profile plot of Drosophila melanogaster (Dm) CENP-C using the DISOPRED3 disorder prediction method at the PSIPRED Protein Sequence Analysis Workbench[55] (http://bioinf.cs.ucl.ac.uk/psipred/). **(c)** Schematic of the Drosophila kinetochore and the experimental design showing CENP-C-based force sensors placed internally (experimental, force applied) versus at the C terminus (control, no force applied).

mitotic mammalian cells measured stall forces of end-on attached kinetochores at ~2–10 pN per kinetochore (below 1 pN per MT)[32]. Thus, existing estimates of end-on-attached kinetochore forces in animals span two orders of magnitude.

*Drosophila* S2 cells are an excellent model system to study kinetochore forces as sister kinetochores are generally under equal opposing forces because bioriented chromosomes do not oscillate. Furthermore, most other well-studied kinetochores possess multiple linkages between the DNA and the outer kinetochore, but the *Drosophila* kinetochore appears to possess a single linker molecule—CENP-C[33]. In this study we aimed to address the substantial inconsistencies in kinetochore force estimates by inserting Förster resonance energy transfer (FRET)- and talin-vinculin-based force sensors into CENP-C since it is ideally positioned as a force-transducing kinetochore component in *Drosophila*. Live-cell measurements of the two reporters yielded comparable estimates (~1–2 pN) of the average amount of force applied per CENP-C reporter molecule at metaphase. MT dynamics, but not dynein, contributed to force production within the measurable range of the FRET-based reporter at bioriented kinetochores. We conclude that metaphase k-fibres exert hundreds of pNs and posit that depolymerizing kt–MT plus-ends are the dominant poleward-directed force producers at bioriented *Drosophila* kinetochores. The findings have implications for understanding the fundamental interplay between force and kinetochore structure, function and evolution.

## Results

**Measuring kinetochore forces with a FRET-based sensor.** CENP-C constitutively localizes to the centromere throughout the cell cycle. It associates with centromeric DNA through its C terminus[34–36] and its N terminus binds directly to the outer kinetochore[37,38] while the rest of the protein, especially in *Drosophila*, is predicted to be highly disordered (Fig. 1a,b). The behaviour of force reporters inserted into the central unstructured region of CENP-C were compared with negative controls in which the sensors were placed at the C terminus so that they would not be subjected to force (Fig. 1c). Stable cell lines expressing each of the force reporters were used in all the experiments and there were no evident dominant negative effects to engineering CENP-C in the manners described.

The first CENP-C force reporter was built using a modified version of a spider-silk-based FRET sensor called TSMod (tension sensor module)[39], comprised of the FRET pair mTurquoise2 and mVenus flanking an elastic linker (Fig. 2a). The application of force to TSMod leads to reduced FRET and, notably, the FRET sensor has been calibrated and shown to report on forces in the 1–6 pN range. Insertion of the TSMod reporter into CENP-C did not evidently disrupt its function as chromosome alignment was indistinguishable between control cells and cells treated with dsRNA targeting the 3′-untranslated region (3′-UTR) of the CENP-C transcript to knockdown (with ~50–60% efficiency) the endogenous CENP-C (Fig. 2b). CENP-C-TSMod exhibited a

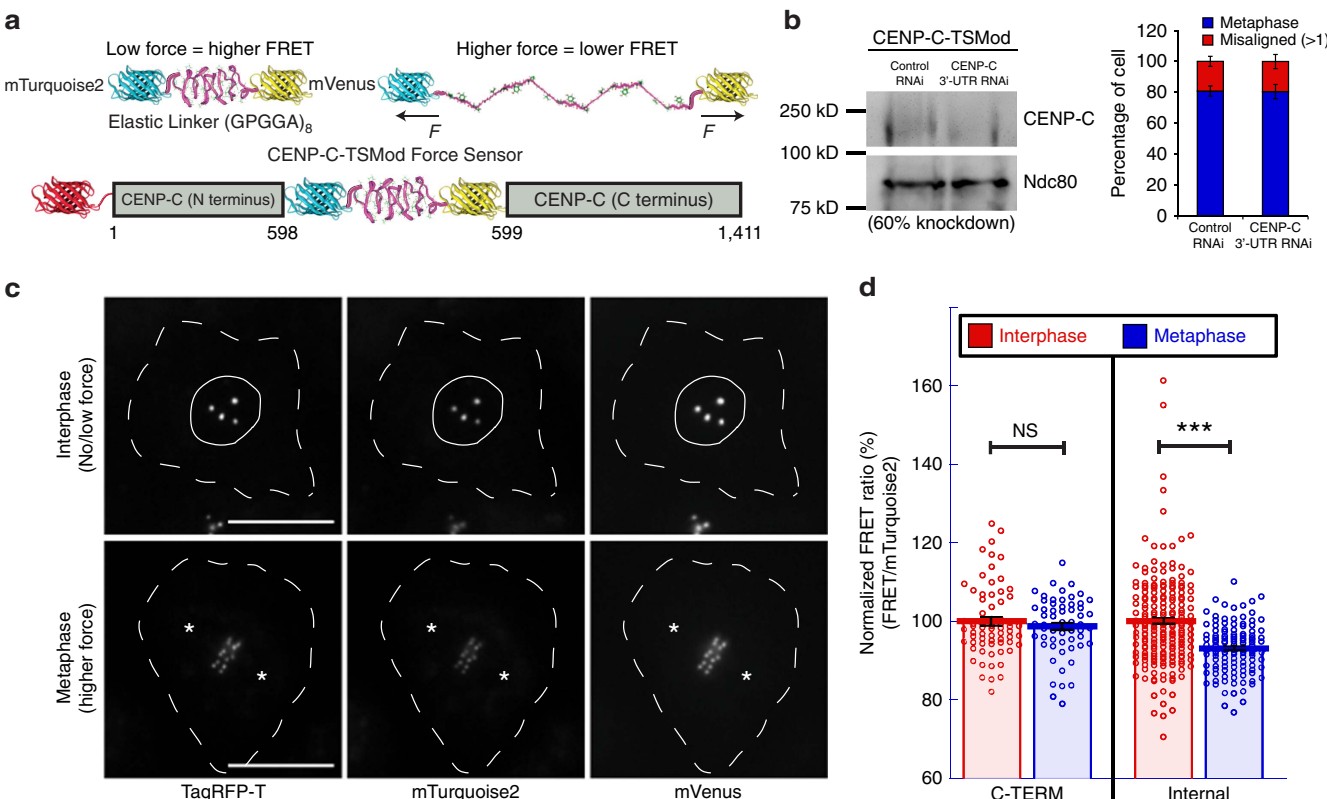

**Figure 2 | Characterization of the CENP-C-TSMod reporter and FRET emission ratio measurements.** (**a**) The TSMod sensor inserted into the middle of TagRFP-T-CENP-C. (**b**) Quantification of chromosome alignment in control and endogenous CENP-C depleted, internal CENP-C-TSMod expressing cells treated with MG132 (representative western blot—left panel). Mean values from two independent experiments; Control RNAi; $n = 105$ cells, CENP-C 3′-UTR RNAi; $n = 104$ cells. (**c**) Cellular localization of the internal CENP-C-TSMod. Dashed lines denote cell boundaries, a solid line outlines the nucleus, and asterisks mark spindle poles. Comparable channels are displayed with identical contrast and brightness scaling. (**d**) FRET emission ratios of the internal and C-terminal reporters normalized to the interphase condition (set to 100%) for each reporter. C-terminal data are from two independent experiments; $n = 67$ interphase cells, $n = 60$ metaphase cells. Internal data are from three independent experiments; $n = 198$ interphase cells, $n = 124$ metaphase cells. Scale bar is 10 μm. Error bars are s.e.m. P values from Mann–Whitney Wilcoxon t-tests are reported: not significant (NS) P value > 0.05, *** P value < 0.0005.

normal localization pattern as it constitutively associated with centromeres throughout the cell cycle when expressed in S2 cells (Fig. 2c). Interphase cells, which do not have assembled kinetochores, were used as the no/low force condition while bioriented metaphase kinetochores were the higher force condition. First, the FRET emission ratio (FRET signal/donor (mTurquoise2) signal) of the reporter was measured. A statistically significant 7% decrease (P value < 0.005) in the FRET emission ratio was measured in metaphase compared with interphase cells expressing the internal TSMod (Fig. 2d; Supplementary Fig. 1a). The metaphase reduction in FRET was not a result of cell cycle effects on the behaviour of the reporter and was dependent on the internal positioning of the sensor as there was not a statistically significant difference (P value > 0.05) in the FRET emission ratio of the C-terminal TSMod in interphase versus metaphase cells.

While the reduction in FRET emission ratio indicated that the internally positioned TSMod was under greater tension in metaphase than in interphase, it was not feasible to estimate the magnitude of force applied to CENP-C from the change in the emission ratio as the reporter was originally calibrated based on changes in FRET efficiency. To overcome this limitation, acceptor photobleaching was applied to measure the FRET efficiency of the CENP-C TSMod reporters (Fig. 3a). In this approach, the FRET efficiency is measured by quantifying the extent to which the fluorescence intensity of the donor (mTurquoise2) increases following photobleaching of the acceptor (mVenus). In general agreement with the FRET emission ratio measurements, a statistically significant ~12.5% decrease (P value < 0.005) in the FRET efficiency of the internal TSMod reporter was measured in metaphase (20.7 ± 0.5% (s.e.m.) FRET efficiency) compared with interphase (23.6 ± 0.6% FRET efficiency) while there was not a statistically significant (P value > 0.05) change in the FRET efficiency of the C-terminal reporter (Fig. 3b; Supplementary Fig. 1b). Chromosome biorientation was required to generate tension as the FRET efficiency of the internal TSMod at kinetochores associated with monopolar spindles following depletion of kinesin-5 was indistinguishable from interphase measurements (Fig. 3b). Importantly, there was negligible (< 0.5%) or no detectable bleed-through from the TagRFP signal into any of the relevant channels used in the FRET-based imaging approaches (Supplementary Fig. 1c,d). The live-cell FRET efficiency measurements compared remarkably well with the FRET efficiency–force estimation of the TSMod developed from single molecule calibrations[39]. Based on the published force estimation curve, the CENP-C-TSMod data are consistent with each CENP-C molecule experiencing forces below the limit of detection in interphase and, on average, ~1.2–1.4 pN at bioriented kinetochores (Fig. 3c).

**MT dynamics contributes to metaphase force generation.** The MT-stabilizing drug taxol was next used to assess the contribution of MT dynamics to kinetochore forces. Addition of greater than ~20–50 nm taxol to *Drosophila* S2 cells results in monopolar spindles, but bipolar spindles form more frequently in the presence of higher concentrations of taxol following depletion of the minus-end directed motor dynein[40]. Dynein heavy chain (Dhc) RNAi treatment exhibited the expected phenotype of a significant increase in metaphase cells. While normal metaphase forces were measured at bioriented kinetochores in DMSO-treated cells following Dhc RNAi, addition of 500 nM taxol resulted in a statistically significant (P value < 0.05) increase in the FRET efficiency of the internal CENP-C-TSMod reporter at bioriented kinetochores in

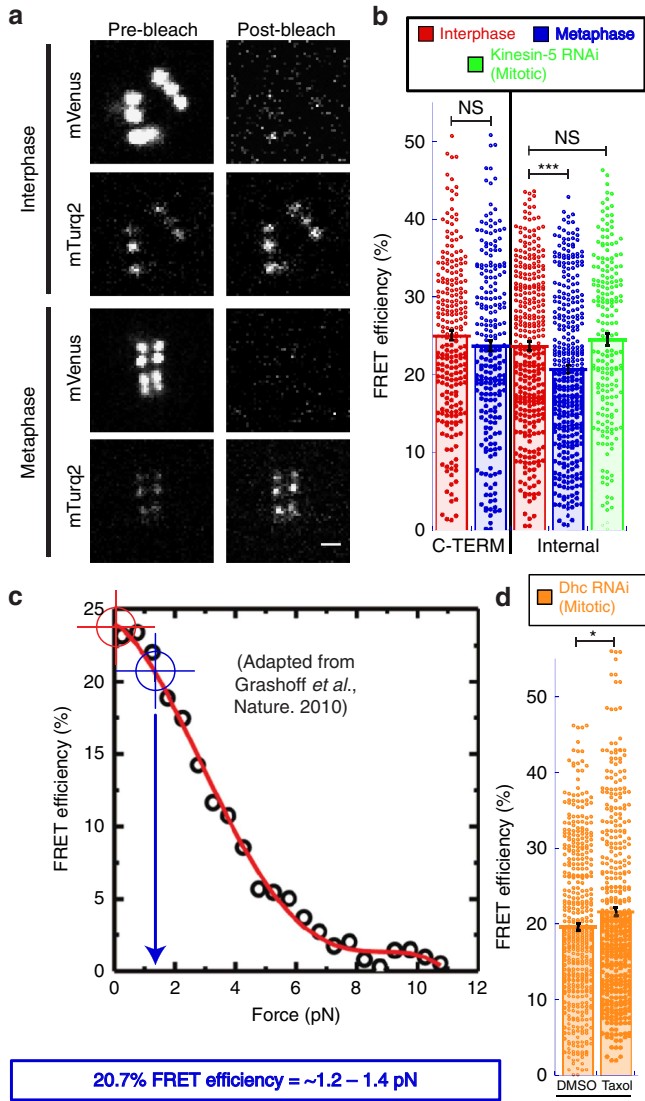

**Figure 3 | Acceptor photobleaching of CENP-C-TSMod reporters and force estimate.** (**a**) Representative examples of acceptor photobleaching of the internal TSMod. Comparable channels are displayed with identical contrast and brightness scaling. (**b**) FRET efficiency measurements from acceptor photobleaching experiments. C-terminal data are from three independent experiments; n = 238 interphase centromeres, n = 229 metaphase centromeres/kinetochores. Internal data are from four independent experiments; n = 318 interphase centromeres, n = 346 metaphase centromeres/kinetochores for untreated cells and two independent experiments for kinesin-5 RNAi-treated cells; n = 198 metaphase centromeres/kinetochores. (**c**) Based on the published calibration of the TSMod reporter[39], a 20.7% FRET efficiency of the internal TSMod reporter corresponds to the application of ~1.2–1.4 pN. The red and blue crosshairs denote the mean FRET efficiency values of the internal reporter in interphase and metaphase respectively. (**d**) FRET efficiency measurements from acceptor photobleaching experiments. Dhc RNAi data are from five independent experiments; n = 454 bioriented kinetochores in DMSO-treated cells, n = 433 bioriented kinetochores in taxol-treated cells. Scale bar, 1 μm. Error bars are s.e.m. Two-tailed P value of Student's t-test is reported for the C-terminal data in **b** otherwise P values from Mann–Whitney Wilcoxon t-tests are reported: not significant (NS) P value > 0.05, *P value < 0.05, ***P value < 0.0005.

Dhc-depleted cells (Fig. 3d). Thus, MT dynamics, but not the minus-end-directed motor protein dynein measurably contribute to force production, within the range of detection of the internal CENP-C-TSMod reporter, at bioriented kinetochores.

**Measuring kinetochore forces with focal adhesion components.** To corroborate the FRET-based measurements, a second CENP-C force reporter was designed based on the focal adhesion protein talin and its binding partner vinculin. Single molecule experiments with the talin rod (TR) domain demonstrated that the application of force to the molecule increased the number of associated vinculin head (VH) molecules purportedly by exposing vinculin binding sites in the TR domain (Fig. 4a)[41]. As with the TSMod reporter, the TR domain was inserted into either the middle of TagRFP-T-CENP-C to measure force or at the C terminus to serve as a negative control (Fig. 1c). Cell lines were built that expressed both VH-EGFP and the CENP-C-TR force sensors based on the hypothesis that the application of force to the TR domain in CENP-C should increase the number of bound VH molecules (Fig. 4b). As observed for the TSMod force sensor, insertion of the TR reporter into the middle of CENP-C did not disrupt its function as chromosome alignment was normal in CENP-C-TR-expressing cells depleted of endogenous CENP-C

(Fig. 4c). Importantly, VH-EGFP localized to kinetochores in a CENP-C-TR-dependent manner (Fig. 4d).

To quantify the number of VH molecules associated per TR domain, a fluorescence correction ratio was determined before imaging the experimental conditions by imaging a reference CENP-C protein with an equal number of EGFP and TagRFP-T fluorophores (Supplementary Fig. 2). The correction ratio was then applied to the measured fluorescence intensities of VH-EGFP and Tag-RFP-T-tagged CENP-C-TR, which were imaged using the identical imaging parameters as the reference conditions. Since the assay involved a two-component system and the VH-EGFP was distributed through the cytoplasm, nucleoplasm and centromeres during interphase, the low/no force condition was created by generating unattached kinetochores through treatment with the MT-depolymerizing agent colchicine. The internal TR-reporter-associated with an average of $\sim 0.9 \pm 0.02$ VH-EGFP molecules in colchicine-treated cells. There was not a significant difference ($P$ value $> 0.05$) between VH molecules bound per TR in colchicine-treated and metaphase cells expressing the C-terminal reporter although the mean number of VH molecules per TR was lower ($\sim 0.75 \pm 0.03$) for the C-terminal reporter. In metaphase cells expressing the internal TR, the VH-EGFP signal at bioriented kinetochores was brighter than at unattached kinetochores with

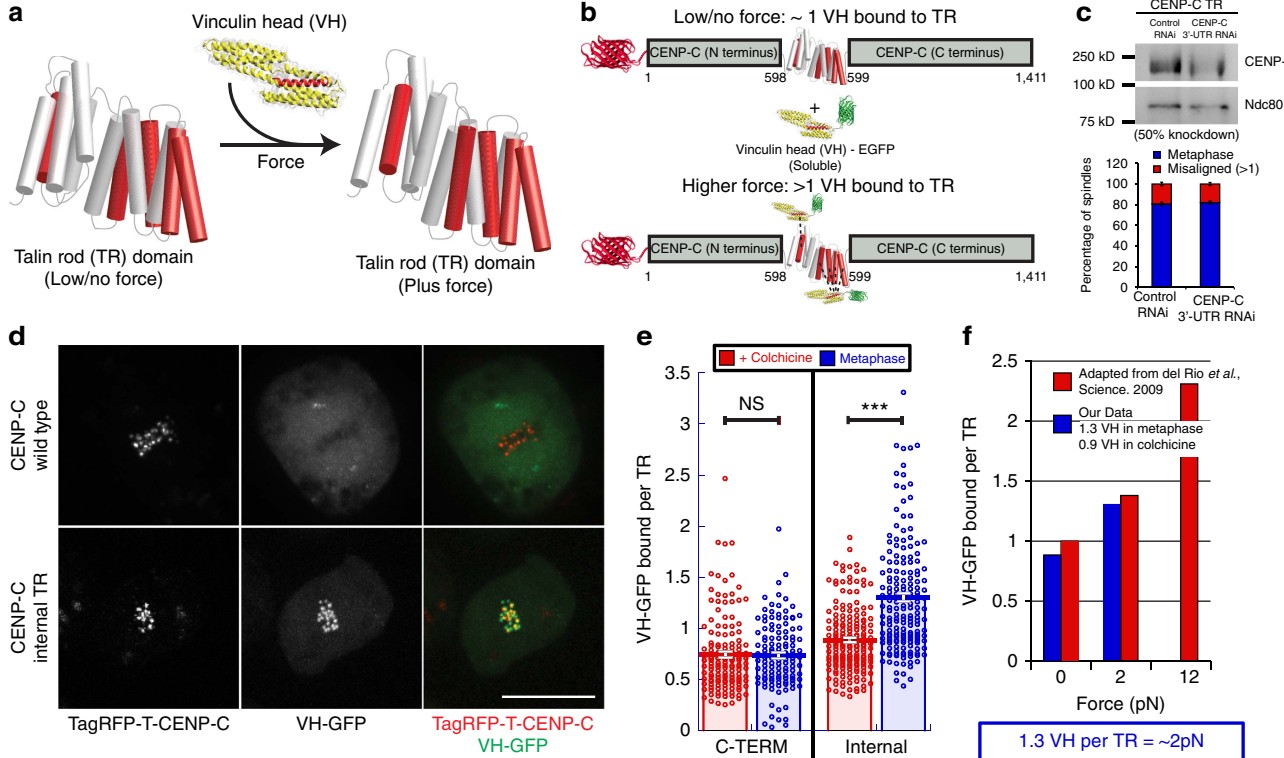

**Figure 4 | Measurements of the CENP-C-TR reporters.** (**a**) Schematic of the proposed effects of force on the structural organization of the TR domain and the potential exposure of VH-binding sites in the TR domain based on a theoretical model of the chicken talin rod domain used here (aa 482–889)[56]. The red regions represent VH-binding domains in TR and the TR-binding domain in VH. (**b**) Experimental design of the internal CENP-C-TR reporter in cells co-expressing soluble VH-EGFP. The dashed lines represent potential VH–TR interactions. (**c**) Quantification of the percentage of MG132-treated cells with metaphase plates and >1 misaligned chromosomes in control and endogenous CENP-C depleted internal CENP-C-TR expressing cells (representative western blot—upper panel). Mean values from three independent experiments; Control RNAi; $n = 162$ cells, CENP-C 3'-UTR RNAi; $n = 162$ cells. (**d**) VH-EGFP localizes to kinetochores in cells expressing TagRFP-T-CENP-C-TR but not wild-type TagRFP-T-CENP-C. (**e**) Counting the number of VH-EGFP molecules associated per TR domain in the internal and C-terminal TR reporters in colchicine-treated and metaphase cells. The C-terminal TR data are from five independent experiments; $n = 146$ colchicine-treated cells, $n = 125$ metaphase cells. The internal TR data are from six independent experiments; $n = 182$ colchicine-treated cells, $n = 179$ metaphase cells. (**f**) Force estimation of the internal CENP-C-TR reporter based on published VH counting from single TR molecule magnetic trap experiments[41]. Scale bar, 10 μm. Error bars are s.e.m. $P$ values from Mann–Whitney Wilcoxon $t$-tests are reported: not significant (NS) $P$ value $> 0.05$, ***$P$ value $< 0.0005$.

an average of $\sim 1.3 \pm 0.05$ VH molecules per CENP-C-TR (Fig. 4e; Supplementary Fig. 3a). Fluorescence measurements were not impacted by the fact that TagRFP-T and EGFP is a FRET pair as FRET was not detectable in either metaphase or colchicine-treated cells (Supplementary Fig. 3b–d).

In prior work, the number of VH molecules per TR domain was counted in the absence of force and after applying 2 or 12 pN to single TR molecules with a magnetic trap[41]. While these measurements were done *in vitro* rather than in living cells, we feel that the approach employed here is bolstered by the fact that a comparable number of VHs per TR was measured *in vitro* in the absence of applied force and in colchicine-treated cells expressing the internal TR reporter (Fig. 4f). The measurement of $\sim 1.3$ VH molecule bound per internal TR domain in living cells is slightly below the average number of VH molecules bound to a TR domain under 2 pN of force applied *in vitro* (Fig. 4f).

## Discussion

The TSMod- and TR-based reporters indicated that each CENP-C molecule experiences, on average, $\sim 1$–$2$ pN of force at bioriented kinetochores. It is important to recognize that both the FRET efficiency measurements and VH counting numbers come from ensembles of molecules some of which are likely in a resting state and others that are in a tense state. Furthermore, since all the measurements are done in the presence of endogenous CENP-C it is possible that the load could be shared unequally between the tagged and untagged CENP-C molecules or distributed through a greater number of CENP-C molecules than are typically present. Thus, the 1–2 pN per CENP-C estimate by no means excludes that forces are differentially distributed through individual CENP-C linkages, and may even underestimate the magnitude of force that can be applied to CENP-C. Nevertheless, the average estimate provides a solid framework from which to parlay our experimental findings into a reasonable proposal for a physiologically relevant range of forces produced by individual kt–MTs and the k-fibre as a whole.

Our MT and k-fibre estimates are based upon a generally accepted concept of the structural organization of the kinetochore (reviewed in Rago and Cheeseman[42]), measurements of the number of CENP-C molecules[43–45] and kt–MTs[46] per *Drosophila* kinetochore, and the force per CENP-C molecule measured here (Fig. 5). The upper and lower limits of the proposal are defined by the range of force per CENP-C (1–2 pN) and previously reported *Drosophila* CENP-C counting experiments[43–45] based on a fluorescent standard (12–31 per MT). In *Drosophila*, we posit that the force produced by a single MT at mature bioriented attachments is distributed between CENP-C molecules arranged as a set of parallel springs that connect to the MT through the Mis12 complex and the Ndc80 complex. It would therefore hold that the force a single MT generates would be the number of CENP-C molecules (12–31) multiplied by the force applied to each CENP-C molecule (1–2 pN) meaning that, on average, a kt–MT exerts $\sim 12$–$62$ pN of poleward-directed force. The total force applied to the kinetochore would equal the force per MT multiplied by the number of MTs in the k-fibre. Since there is an average of 11 microtubules bound to *Drosophila* S2 cell kinetochores[46], we propose that a typical bioriented kinetochore in these metazoan cells experiences between $\sim 135$ and 680 pN of poleward-directed pulling forces. This magnitude of force differs significantly from that measured in live-cell optical trapping experiments[32], but largely agrees with the classic cell-based microneedle experiments in insect cells[23,27] as well as with the force estimate from a recent study combining experimentation and computational theory in mammalian cells[47].

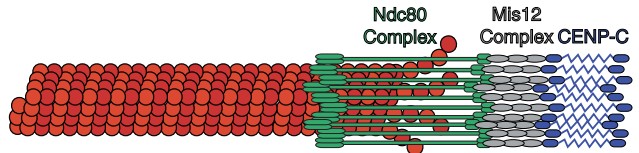

A simple *Drosophila* kinetochore model: Linkages arranged as a set of parallel springs

$$F_{MT} = F_{CENP\text{-}C} \times \text{\# CENP-C molecules per MT}$$

$$F_{kt} = F_{MT} \times \text{\# kinetochore-microtubules}$$

| | $F_{CENP\text{-}C}$ | # CENP-C per MT | $F_{MT}$ | # kt-MTs[+] | Max $F_{kt}$ |
|---|---|---|---|---|---|
| Low | 1 pN | 12.3* | 12.3 pN | 11 | 135 pN |
| High | 2 pN | 30.8** | 61.5 pN | 11 | 677 pN |

**Figure 5 | An estimate of the magnitude of force applied to bioriented kinetochores.** The *Drosophila* kinetochore is envisioned as linkages between the DNA and the kt–MT arranged as a set of parallel springs that distribute forces produced by MT dynamics through them. In *Drosophila*, the known path from the DNA to the MT is CENP-C, the Mis12 complex, and the MT-associating Ndc80 complex. For simplicity, *Dm*Spc105/KNL-1 has been omitted. The TSMod- and TR-based force sensors reported that each CENP-C molecule experiences, on average, $\sim 1$–$2$ pN of force. If the poleward force produced by a kt–MT is equally distributed across the CENP-C linkages then, based on CENP-C counting experiments, each kt–MT could generate between $\sim 12$ and 62 pN. Since there are 11 MTs bound to S2 cell kinetochores, if each kt–MT applies maximum poleward force then the total force applied to the kinetochore would be as high as 135–677 pN. *Measured by Schittenhelm et al.[45] based on Cse4 standards from Joglekar et al.[43] **Adjusted 2.5-fold based on recalibrated fluorescence measurements by Lawrimore et al.[44] [+]Measured in *Drosophila* S2 cells by Maiato et al.[46]

We favor the interpretation that the depolymerizing kt–MT plus-end is the dominant poleward-directed force producer at bioriented *Drosophila* kinetochores for several reasons: (1) our estimate of up to 62 pN per kt–MT nearly equals measurements of the maximum amount of force depolymerizing MT plus-ends generate *in vitro*[10], (2) suppressing plus-end MT dynamics with 500 nM taxol reduces the amount of force at bioriented kinetochores. Prior work concluded that 1 µM taxol was necessary to fully suppress MT dynamics in S2 cells[40]; unfortunately, sufficient numbers of bioriented kinetochores could not be measured in 1 µM taxol. While the data indicate that dynein is not a major force generator at metaphase kinetochores, regulators aside from MTs may contribute since 500 nM taxol did not reduce the force to interphase levels. We cannot exclude the possibility; however, that 500 nM taxol dampens but does not fully suppress MT dynamics. Interestingly, since kinetochore/centromere associated kinesin-13 family members are capable of depolymerizing taxol-stabilized MT ends[48], it would be worthwhile to further investigate the contribution of these motors to force generation at bioriented kinetochores. It is also noteworthy that just 10 nM taxol is sufficient to reduce kt–MT flux by >90% in S2 cells[49] and so our data do not rule out k-fibre flux as a contributor to metaphase force generation. Given that k-fibres flux and pull, and that polymerizing and depolymerizing MTs are present within the same kinetochore[50,51], the relative contributions of kt–MT polymerization versus plus (and perhaps minus)-end depolymerization to kinetochore force transduction remains an open and important question, which has recently been investigated using FRET-based Ndc80 reporters in budding yeast[52].

We hypothesize that kt–MT plus-end depolymerization sets the maximum force applied to bioriented kinetochores and that

this limit must be highly conserved since it is derived, at its most fundamental level, from GTP hydrolysis[11,53]. Thus, kinetochores likely emerged and evolved in the presence of an evolutionarily fixed maximum force generator—the depolymerizing kt–MT. A common evolutionary strategy to building a kinetochore appears to have involved distributing k-fibre forces through a sufficient number of linkages such that individual components are not subjected to very high forces that would denature them or damage the underlying DNA. Defining the physical properties of force-transducing kinetochore components as well as how forces are transmitted across them will provide fundamental mechanical insights into kinetochore function, and cell division in general.

Why would cell division in metazoan cells benefit from having k-fibres capable of applying hundreds of pNs to kinetochores when much lower forces are sufficient to move chromosomes? It is possible that the ability of k-fibres to produce high forces allows for proper resolution of merotelic attachments in anaphase[54] or for segregating chromatids to plow through unexpected obstacles that may arise. However, because the measurements here are in close agreement with Nicklas' estimates of the amount of force (30 pN per MT) required to stabilize kt–MT attachments[27], we propose that the application of hundreds of pNs to kinetochores is most critical before anaphase to stabilize the geometrical configuration with the best segregation prospects—bioriented attachments.

## Methods

**Drosophila S2 cell lines.** All cell lines were cultured in Schneider's (Life Technologies) media supplemented with 10% heat-inactivated fetal bovine serum and $0.5 \times$ antibiotic-antimycotic cocktail (Life Technologies), maintained at 25 °C. Transgenic cell lines were generated by transfecting DNA constructs using the Effectene Transfection Reagent system (Qiagen), following manufacturer protocol. Protein expression was confirmed by fluorescence microscopy. Cells were split in the presence of Blasticidin S HCl (Fisher) and/or Hygromycin (Sigma) to select for expressing cells.

**DNA constructs.** The pMT-TagRFP-T-CENP-C construct was generated in multiple steps: (1) CENP-C was amplified from the cDNA, with 5′ SpeI and 3′ SacII sites and inserted into pMT-V5 B vector; (2) endogenous CENP-C promoter was amplified from genomic DNA with 5′ XbaI site and 3′ KpnI site and inserted into the above plasmid purified from dam-/dcm- *Escherichia coli*; (3) TagRFP-T was amplified with 5′KpnI and 3′SpeI sites, and inserted between the promoter and CENP-C. Then, CENP-Cprom-TagRFP-T-CENP-C was amplified from the above construct with flanking XbaI sites and inserted into pMT-V5 B vector purified from dam-/dcm- *E. coli* to generate pMT-CENP-Cprom-TagRFP-T-CENP-C. To generate the tension sensor module (TSMod) construct, mTurquoise2 and mVenus were first inserted into pMT-V5 B vector between KpnI and SpeI sites, and NotI and SacII sites, respectively to build pMT-mTurquoise2-mVenus. The spider silk DNA sequence with flanking homology regions to mTurquoise2/SpeI site (5′ end) and NotI/mVenus (3′ end) was synthesized (Life Technologies) and inserted between mTurquoise2 and mVenus in the pMT-mTurquoise2-mVenus plasmid by Gibson assembly. To generate the CENP-C-TSMod construct, the TSMod was amplified from the above construct with primers upstream of mTurquoise2 and downstream of mVenus with flanking ClaI sites, and inserted into pMT-CENP-Cprom-TagRFP-T-CENP-C purified from dam-/dcm- *E. coli*. To generate the control construct with the tension sensor in the C terminus of CENP-C, a ClaI site was engineered into a pMT-CENP-Cprom-TagRFP-T-CENP-C (no stop codon) construct by inserting annealed complementary oligos encoding a ClaI site flanked by XbaI sites downstream of CENP-C at the XbaI site and the TSMod was then inserted into this newly engineered ClaI site as the internal ClaI site in CENP-C was methylated.

The CENP-C-TR construct was generated by amplifying talin rod domain (aa 482–889) from chicken gizzard cDNA (Zyagen) with flanking ClaI sites, and inserted into the pMT-CenpCprom-TagRFP-T-CENP-C construct described above. To generate the control construct, in which talin rod was at the 3′ end of CENP-C, the rod domain was amplified with flanking XbaI sites and inserted downstream of CENP-C.

To generate the VH-GFP construct, the vinculin head domain (aa 1–258) was PCR amplified from a vinculin-mVenus construct [a gift from Martin Schwartz (Addgene plasmid # 27300)] with 5′ KpnI and 3′ XbaI sites and inserted into the pMT-V5 B vector with the EGFP sequence between the XbaI and SacII sites. The CENP-C promoter was then inserted upstream of the vinculin gene with flanking KpnI sites to drive expression.

**Double-stranded RNA production.** DNA templates for Dhc64C (CG7507), KLP61F (CG9191), and CENP-C 3′-UTR (CG31258) were produced to contain $\sim 500$ bp of complementary sequence flanked by T7 promoter sequence. Double-stranded RNAs (dsRNAs) were synthesized overnight at 37 °C from the DNA templates using the T7 RiboMax Express Large Scale RNA Production System (Promega) following manufacturer protocol. For RNAi, media was aspired off semi-adhered cells at 25% confluence, replaced with 1 ml of serum-free Schneider's medium containing 20 μg of dsRNA, and after 1 h, 1 ml of fresh Schneider's plus FBS was added to the wells and incubated for 2 (Dhc, Klp61F) or 4 (CENP-C) days at 25 °C.

**Western blot.** A total of 20 μg of protein was loaded onto a 10% SDS–PAGE gel, run out, and transferred to a nitrocellulose membrane on the Trans-Blot Turbo transfer system (Bio-Rad Laboratories) for 15 min. All antibodies were diluted in TBS with 0.1% Tween and 5% milk. The membrane was first incubated with anti-CENP-C serum (gift from Bibi Mellone) at 1:7,500, followed by anti-Ndc80 antibody (made in house) at 1:5,000 as a loading control. Guinea pig (703-035-155) and chicken (706-035-148) HRP secondary antibodies (Jackson ImmunoResearch Laboratories), diluted at 1:5,000, were used in conjunction with their respective primaries and imaged with a GBox system controlled by GeneSnap software (Syngene). ImageJ was used to measure band intensities and the CENP-C signal was normalized to the Ndc80 loading control to determine the knockdown efficiency. Uncropped images of the western blots are shown in Supplementary Fig. 4.

**FRET ratio imaging and analysis.** Cells were allowed to adhere to acid-washed, concanavalin A (Sigma-Aldrich) coated coverslip (Corning) for exactly 1 h, then assembled into a rose chamber containing Schneider's media with drugs or solvent control, when appropriate and subjected to imaging at 25 °C. Cells were imaged for a maximum of 1 h on a TiE inverted microscope (Nikon) equipped with an iXON EMCCD camera (Andor Technology) using a $100 \times 1.4$ numerical aperture Plan Apo violet-corrected series differential interference contrast objective (Nikon). Metamorph software (Molecular Devices) was used to control the imaging system. For imaging the TagRFP-T-CENP-C-TSMod FRET reporter, mitotic cells were identified by the presence of paired sister centromeres and the absence of a nucleus, and the best focal plane was determined in the RFP channel. Sequential images of mTurquoise2, mVenus and FRET were taken with equal exposure times. Background-corrected fluorescence intensities for mTurquoise2 and FRET were measured in Metamorph software using region-in-a-region background subtraction by drawing concentric larger and smaller regions manually in MetaMorph around clusters of kinetochores/centromeres that were in focus. The reported FRET emission ratios represent the ratios of the background corrected FRET over the background corrected mTurquoise2 total intensities. The following equations were used:

$$\text{Background signal} = \frac{\text{Integrated fluorescence intensity}_{\text{larger area}} - \text{Integrated fluorescence intensity}_{\text{smaller area}}}{\text{Larger Area} - \text{Smaller Area}}$$

(1)

$$\text{Total Intensity} = \text{Integrated fluorescence intensity}_{\text{smaller area}} - (\text{Background signal} \times \text{Smaller Area})$$

(2)

Bleed-through of the TagRFP-T into the CFP, YFP and FRET channels was measured by imaging TagRFP-T-α-tubulin-expressing cells under identical conditions as the FRET imaging experiments. Background corrected CFP, YFP, and FRET signals in MT-containing regions were then ratioed to the TagRFP signal from that region.

**Acceptor photobleaching FRET.** Cells were seeded onto a Concanavalin A-coated acid-washed coverslip and allowed to adhere for 1 h. For Dhc RNAi experiments, cells were incubated for 1 h in taxol or 0.1% DMSO after allowing them to adhere to the coverslips for 1 h and imaging was done for a maximum of 60 min following the 1 h drug or DMSO treatment. Coverslips were then assembled into a rose chamber containing Schneider's media (containing 500 nM taxol or 0.1% DMSO where appropriate) and imaged at 25 °C. All images were collected using a TiE inverted microscope (Nikon) coupled with an A1R laser scanning confocal system (Nikon) using a $60 \times 1.4$ NA Plan Apo objective (Nikon). Elements (Nikon) was used to control the imaging system. Best focal plane was determined by taking a single image using the 561 nm laser. Single plane images were acquired using the 445 and 514 nm lasers pre- and post-photobleaching. To photobleach the acceptor fluorophore, a square region was drawn around a metaphase plate or interphase centromeres, and photobleached with the 514 nm laser, using a 50 mw laser at 55% laser power. All image quantifications were done using Fiji Image J software. A region was drawn around individual kinetochores/centromeres in pre- and post- photobleach images to obtain the mTurquoise2 (donor) integrated intensity, which was corrected by subtracting the background signal obtained by placing the same sized region in the cytoplasm/nucleoplasm of the same cell. The following equation was applied to obtain the FRET efficiency:

$$\text{FRET}_{\text{eff}} = 1 - \frac{\text{Intensity}_{\text{donor(pre-bleach)}}}{\text{Intensity}_{\text{donor(post-bleach)}}}$$

All statistical analyses were performed using R or Prism. While mTFP1 was replaced with mTurquoise2 in the TSMod reporter used in this study, we feel that it

is appropriate to present a force estimate from our reporter that is based on the theoretical calibration of the original TSMod[36] because the two reporters exhibit identical zero-force FRET efficiencies and have comparable Förster radii. Bleed-through of the TagRFP-T into the CFP and YFP channels was measured by imaging TagRFP-T-α-tubulin-expressing cells under identical conditions as the acceptor photobleaching experiments. Background corrected CFP and YFP signals in MT-containing regions were then ratioed to the TagRFP signal from that region.

**Talin rod—vinculin imaging and analysis.** Cells co-expressing CENP-C-TR and VH-EGP were seeded near full confluency in 500 μl volume onto concanavalin A coated acid-washed coverslips. After 20 min the volume was brought up to 2 mls with fresh Schneider's media ( + 0.1% DMSO or 25 μM colchicine) and assembled into a rose chamber ~ 40 min after seeding. For colchicine-treatments, the cells were treated with 25 μM colchicine for 60 min before seeding them onto coverslips. The cells were then imaged on the microscope described above (in 'FRET ratio imaging and analysis') between 45 and 90 min post seeding. Mitotic cells were identified as described above and the best focal plane was determined in the RFP channel and sequential images of TagRFP-T and EGFP were taken with equal exposure times. Background-corrected fluorescence intensities for TagRFP-T and EGFP were measured using region-in-a-region background subtraction as described above. To obtain the correction ratio, cells expressing TagRFP-T-CENP-C-EGFP and treated with 0.1% DMSO were imaged before each experiment using identical imaging conditions as would be applied to the experimental conditions. Region-in-a-region background subtraction was applied to measure the ratio of GFP to RFP signal intensities (the GFP signal was typically ~ 3.5 × greater than that of RFP under the imaging conditions). Since TagRFP-T-CENP-C-EGFP has equal numbers of EGFP and TagRFP-T molecules, the correction ratio determined for that day was used to determine the number of VH molecules per TR in the experimental conditions by dividing the background corrected VH-EGFP to TagRFP-T-CENPC-TR ratio by the correction ratio. For example, if the correction ratio for a given day was measured to be 3.5 and the (background corrected) ratio of VH-EGFP to TagRFP-T-CENP-C-TR kinetochore signals was measured to be 7 then the number of VH per TR would be 2. This method does not count the total number of TRs or VHs per kinetochore but rather the number of VH molecules per CENP-C-TR.

To investigate if FRET was occurring between TagRFP-T tagged CENP-C-TR and VH-EGFP, cells were imaged on an Eclipse Ti-E inverted microscope (Nikon) equipped with a Borealis (Andor) retrofitted CSU-10 (Yokogawa) spinning disk head and ORCA-Flash4.0 LT Digital CMOS camera (Hamamatsu) using a 100 × 1.49 numerical aperture Apo differential interference contrast objective (Nikon). Cells expressing either EGFP-α-tubulin or TagRFP-T-α-tubulin were imaged and background corrected signals from EGFP (donor) and TagRFP-T (acceptor) into the FRET channel (GFP excitation, RFP emission) in MT-containing regions were ratioed to the background corrected tubulin signal from that region to define the spectral bleed-through (bt) into the FRET channel (~ 6% for EGFP, and ~ 12% for TagRFP-T). TagRFP-T tagged CENP-C-TR and VH-EGFP expressing cells were then imaged under identical imaging conditions as the tubulin-expressing cells and corrected FRET (cFRET) was determined by subtracting the donor and acceptor bt from the background corrected raw FRET signal using the following equation:

$$cFRET = raw\ FRET - EGFP_{bt} - TagRFPT_{bt}$$

The corrected FRET values were divided by the donor (EGFP) intensity and reported as a cFRET ratio.

**Data availability.** The data that support the findings of this study are available from the corresponding author upon request.

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

## Acknowledgements

Thank you to Barbara Mellone (UConn) for the anti-CENP-C antibody. The authors also thank Tae Yeon Yoo and Dan Needleman (Harvard University) for their assistance with FLIM. We are grateful to Andrea Freikamp and Carsten Grashoff (Max Planck Institute of Biochemistry) for communicating unpublished findings. The acceptor pho-tobleaching data was gathered in the Light Microscopy Core and Nikon Center of Excellence at the Institute for Applied Life Sciences, UMass Amherst with support from the Massachusetts Life Sciences Center. This work was supported by an NIH grant (5 R01 GM107026) to T.J.M. and by Research Grant No. 5-FY13-205 from the March of Dimes Foundation to T.J.M., as well as support from the Charles H. Hood Foundation, Inc., Boston, MA. to T.J.M.

## Author contributions

T.J.M. conceived of the project, carried out the FRET ratio and talin-vinculin experiments and analyses, and wrote the paper. A.A.Y. made the C-terminal CENP-C TSMod construct and conducted the acceptor photobleaching experiments and analyses. S.C. made all other DNA constructs and the cell lines. The manuscript was edited by A.A.Y., S.C. and T.J.M.

## Additional information

**Competing financial interests:** The authors declare no competing financial interests.

