## [Peer Review File · Nature Communications]

Reviewers' comments:

Reviewer #1 (Remarks to the Author):

Summary and general comments:

Mechanical tension at kinetochores is widely believed to provide the cue that dividing cells sense in order to distinguish whether their chromosomes are properly attached to the mitotic spindle. Improper attachments lacking tension are selectively released to give another chance for proper attachments to form. Dividing cells also sense kinetochore tension in order to decide whether to proceed into anaphase, when the chromosomes are split and the replicated sister chromatids segregate to opposite sides in preparation for cytokinesis. Both of these tension-dependent surveillance systems evolved to prevent the creation of daughter cells with the wrong numbers of chromosomes - a condition called aneuploidy that can cause birth defects and is associated with cancer. Because kinetochore tension is so vital for correcting mitotic errors and for controlling cell cycle progression, it is crucial to know the magnitude of these forces *in vivo*. Until the levels of force *in vivo* are known, for example, it is not possible to fully assess whether reconstitutions of spindle activities using purified components have successfully matched the physiological situation. However, measuring mechanical forces inside living cells is extremely challenging. Only a few prior studies have even attempted to do so. The reported force values from these pioneering studies span orders of magnitude, and thus "a surprising lack of consensus" exists about the physiologically relevant force levels (as nicely stated by Ye and co-workers in their manuscript).

Ye, Cane, and Maresca have addressed this deficiency by engineering two artificial, fluorescence-based tension sensors into the kinetochores of *Drosophila* S2 tissue culture cells. One sensor, TSMoD, uses fluorescence resonance energy transfer (FRET) between two fluorescent proteins linked via an elastic polypeptide derived from spider silk. Higher tension stretches the fluorophores apart and thus reduces FRET. Another sensor uses the talin rod (TR) domain, which binds to a vinculin head-GFP fusion (VH-GFP) in a manner that is enhanced by tension applied to TR. For both sensors, the relationship between mechanical tension and fluorescence (either FRET efficiency or GFP intensity) has been directly mapped by prior *in vitro* studies (del Rio 2009, and Grashoff 2010). The TSMoD sensor has also been applied *in vivo* in at least two previous studies, to (i) measure force levels in focal adhesions (Grashoff 2010), and to (ii) assess qualitatively the forces in kinetochores in budding yeast (Suzuki 2016 Nat Cell Biol). However, the present study by Ye and co-workers is perhaps the first to use two different sensors to quantify and validate the force levels on any particular molecule *in vivo*. The use of

two different sensors is a significant strength of this work, giving considerably more confidence, in comparison to previous studies, that accurate forces have been measured.

The sensors were engineered into the kinetochore protein, Cenp-C, which links spindle-binding elements of the outer kinetochore to DNA-binding elements of the inner kinetochore. The primary conclusions are that Cenp-C is normally unloaded in interphase in *Drosophila* S2 cells and then when chromosomes bi-orient, each molecule of Cenp-C carries 1 to 2 pN of tension. This result is interesting because it implies a fairly high level of total tension for each kinetochore and each kinetochore-microtubule attachment site. The exact copy number of Cenp-C molecules per kinetochore is uncertain, so the total tension can only be estimated based on the measured tension per Cenp-C and the approximate numbers of Cenp-C per kinetochore. Nevertheless, the estimated force totals are high, 12 to 60 pN per kinetochore-attached microtubule, and 140 to 680 pN per whole kinetochore.

This paper was a pleasure to read. With a couple of minor exceptions, detailed below, the text is very carefully written and referenced. The data are presented clearly and concisely, and the conclusions seem well-justified. The conclusions are also important and likely to be of interest to a wide audience, not only to mitosis researchers but also to the wider community of biologists interested in mechanically responsive cell systems. While the use of fluorescence-based force sensors *in vivo* is not entirely new, this study establishes a higher bar for how best to use them. By comparing the results from two different sensors, the results are less likely to be contaminated by factors unrelated to mechanical tension. One hopes that other researchers using fluorescence-based force sensors will adopt this same strategy (and that additional sensors will be developed as well).

One important concern is that there does not seem to be any information about whether insertion of the sensors affected kinetochore function. The sensors are fairly large proteins, and they have been inserted into a key element of the kinetochore. Thus it is important to determine whether they have had any detrimental effects on kinetochore function. A sensor that disrupts kinetochore function would not be expected to report on normal levels of kinetochore force. If this concern can be addressed, then the paper seems very well suited for publication in *Nature Communications*. More specific comments are listed below - these are intended to be helpful to the authors as they revise for publication.

Specific comments:

Were the cells with sensor-engineered Cenp-C healthy? Was the duration of mitosis altered relative to normal cells? Was there any change in the fraction of mitotic cells? Did the spindles look normal? Was the kinetochore arrangement normal? For example, were inter-kinetochore distances normal? It seems crucial to provide at least some information about whether the

sensors affected kinetochore function.

Almost no information is given about the structure of Cenp-C. What is known about its domain structure? Does it have domains likely to form coiled-coils or globular folds or anything with recognizable structural features? It would be helpful to include this kind of information in the introduction. How was the precise location for insertion of the sensors chosen? This kind of information would be helpful in the methods, for those who might want to replicate the work in the future.

While the fluorescence images in Figures 2B, 2D, and 3C show distinct kinetochore dots, no other cellular landmarks are visible so it is difficult to put these dots into context. Can you show brightfield images of the S2 cells? Also, it would be helpful to understand the arrangement of spindle microtubules in the metaphase cells, and why the dots are clustered even in the interphase cells (i.e., in Figures 2B and 2D). A general audience and even many mitosis experts will be less familiar with this particular cell type, so it would be useful to help the reader understand the cellular context of the imaged kinetochores.

The use of space in the figures seems imbalanced. The cartoon diagrams of the TSMOD sensor shown in Figures 2A are huge, whereas the fluorescence images in Figure 2B show only very tiny kinetochore dots. Similarly, the diagrams of TR in Figures 3A and 3B are huge, with smaller images in Figure 3C. Why not make the images bigger and the cartoon diagrams smaller? Or else use the extra space freed by smaller diagrams to show more example images? (Or to show brightfield images or images of the microtubules in the same cells, if available - see previous comment.)

The engineered Cenp-C-sensors were expressed (using a Cenp-C promoter) from DNA constructs that were transiently transfected into the cells. Presumably the endogenous, untagged Cenp-C was also present. Could the load be unequally shared between the tagged and untagged Cenp-C molecules? This possibility is apparently not considered.

The methods state simply that "mitotic cells were identified". On what basis?

From page 5 - "Cenp-C-TSMOD localized normally to centromeres throughout the cell cycle when expressed in S2 cells". This sentence leaves ambiguity about the normal pattern of localization of Cenp-C to centromeres. It states clearly enough that the localization pattern of Cenp-C-TSMOD matches that of untagged Cenp-C. However, the sentence does not clearly define whether or not Cenp-C is found at centromeres during all phases of the cell cycle. To help readers less familiar with the temporal patterns of centromere localization, it would be good to be more explicit about what the normal pattern is for Cenp-C.

From page 8 - "in cases where there were only 2 - 4 (rather than 12 - 13) CenpC linkages to the fluxing MT." This sentence is terribly confusing. In what situation would you expect only 2 - 4 linkages? If only 2 - 4 Cenp-C linkages were placed under tension by "a fluxing MT polymerizing against a barrier within the kinetochore", then wouldn't the remaining 8 - 11 Cenp-C molecules be relaxed, and thus the average tension you'd measure with your sensor would be well below 1 - 2 pN? This speculative discussion point seems very unclear and possibly unnecessary. Is there a reference missing here?

"...the observation that Dhc RNAi did not reduce kinetochore tension supports the conclusion that depolymerizing kt-MT plus-ends are the dominant poleward-directed force producers at bioriented Drosophila kinetochores." This seems overstated. The data are consistent with a depolymerization-based mechanism for force production, but other possibilities are not ruled out.

"Given that k-fibers both flux and pull and that straight (presumably polymerizing) and curled (presumably depolymerizing) MTs are present..." It seems you are referring here to curled versus straight protofilaments at the ends of kinetochore-attached microtubules, not microtubules per se. This sentence needs work.

In general, the final paragraph of the discussion is wordy and not very clear. Given the very clear writing in the paper leading up to this final paragraph, the latter stands out.

Please explain more clearly what regions were chosen for fluorescence quantification. "...using a region-in-region background subtraction..." What size were the regions of interest? Did they encompass individual kinetochores, or clusters of many kinetochores? Why not show an example region of interest overlaid onto a fluorescence image, to provide your readers with a clearer idea of what you did?

In Figure 3, structurally detailed views of the talin rod domain under low and high forces are shown. Where do these structures come from? Please provide a reference, or a PDB number, so that readers can understand the basis of these diagrams. The structural changes with force seem very subtle. Is there no unfolding of TR?

It would be useful to put the histograms in Supplementary Figure 1 all on the same axes. Maybe merge the histograms as in Supplementary Figure 3 so the two that are supposed to be compared are on the same graph.

In Figure 3E, the authors' data from only the test condition (with TR sensor inserted into the middle of Cenp-C) are compared to the del Rio 2009 measurements. Why not also indicate the control conditions (with C-terminal TR sensor, and with colchicine) on the same plot? It is

noteworthy that the control conditions suggest slightly less than 1 VH per Cenp-C-TR, which matches nicely with del Rio's measurement of 1 VH per TR under no load.

Reviewer #2 (Remarks to the Author):

Motivation: While mechanics play a critical role in metaphase, the forces involved in the process are poorly defined. Previous studies have attempted to quantify these forces through optical trapping studies in yeast and metazoans, but estimates have fallen across two orders of magnitude.

Summary: Ye, Cane, and Maresca measure the magnitude of force applied to the kinetochore during mitosis using two genetically encoded tension sensors. First they measure tension using Grashoff's TSMOD spider silk FRET sensor inserted into CENP-C. They corroborate their results using a Talin Rod sensor, with vinculin head binding as a readout. Both tension sensors suggest 1-2pN/CENP-C. Combining this value with previously reported values of CENP-C per microtubule and microtubules per kinetochore, they estimate that the each kinetochore experiences 135-680 pN of tension.

Significance: This is a fundamental study that contributes to our understanding of metaphase and molecular mechanics in cell division.

Novelty: This is the first use of genetically encoded tension sensors to quantify the mechanical forces applied to the kinetochore. To our knowledge, this is also the first use of the talin rod domain as a genetically encoded tension sensor. Previously it has only been characterized in single molecule experiments.

Commentary on Significance and Novelty: This paper does not address a new question; several studies have already attempted to quantify kinetochore mechanics. The authors remark that it has been challenging to measure the forces applied to the kinetochore, but they fail to address why these forces have been so hard to quantify and why these discrepancies arise. I would suggest incorporating more discussion to address this point.

Although the use of genetically encoded tension sensors is clearly an improvement, the authors do not include a discussion as to why their measurements are more reliable than previous studies. As mechanobiologists we know the importance of making physiologically relevant

measurements. This should be outlined more clearly for the nonexpert reader. The TSMOD sensor, while not previously used in this context, is not in itself novel. The talin rod domain sensor is novel, however they do not emphasize its novelty and there are some concerns regarding those specific experiments (see below). Finally, this paper ends with the sentence "However, because our measurements are in close agreement with Nicklas' estimates of the amount of force (30 pN/MT) required to stabilize kt-MT attachments, we propose that the application of hundreds of pN to kinetochores is most critical before anaphase to stabilize biooriented attachments." This seems to be an unusual claim since what is proposed has already been proposed by Nicklas' work.

The development of tension sensor probes for the kinetochore presents sufficient novelty to justify why the article belongs in Nature Communications. But technical concerns need to be addressed to ensure the robustness of the conclusions.

General comments:

1. Note that the estimated FRET efficiencies produce an ensemble average FRET efficiency, which likely includes molecules that are at a resting state and molecules that are in a tense state. Therefore, when trying to estimate the amount of applied force it is imperative to emphasize that these are somewhat lower bound estimates of tension rather than average values of tension. For example, it is possible that 10% or 50% of the FRET probes imaged are not experiencing any force, while the remaining probes experience 6 pN of force. Therefore the average would be 2 pN. In effect, one is always estimating a minimum force that each molecule would experience to produce the observed FRET efficiency. This point needs to be clearly indicated.
2. The original TsmOD (Nature 2010) used mTFP1 as the donor fluorescent protein. In this work, the mTurquoise2 donor was used instead. Unfortunately, this fluorophore has different optical properties (Quantum yield, extinction coefficient, lambda max, donor lifetime) and hence the J-integral and the forster distance are different from that of the fluorophores used for the TsmOD. Therefore, the TsmOD calibration can not be used directly to determine the amount of applied force. FLIM was used in the original TsmOD work to determine the QE and the extension of the sensor.
3. Statements that a "statistically significant 7% decrease in FRET emission ratio was measured in metaphase..." need to be backed up with a statement of the p-value and also the standard deviation or the SEM of the mean difference between the control group and the strained group.

FRET sensor:

CENP-C is labeled with a TagRFP protein. The authors do not provide justification for why they labeled this protein for the TSMOD tension sensor, and they only seem to use it to focus when imaging. As there are already two fluorescent proteins present, it was not clear why RFP was expressed with this tension sensor. It likely contributes to unwanted bleedthrough and

background signal in the FRET measurements. The authors should address bleedthrough and FRET concerns to make sure that this third and seemingly unnecessary fluorescent protein does not contaminate their signal.

Figure 2) Figure 2B displays the interphase and metaphase data at very different magnifications, making it hard to compare. These images need to be displayed in a consistent manner such that the magnifications do not obscure the findings. The authors may also consider adding a scale bar or using a color scale to better display these changes in fluorescence. In addition, the images in 2B require further explanation as to what exactly we are looking at in relation to the cell. Perhaps these are questions that a reader within this biological field could answer, however as Nature communications will likely have significant readership that are nonexperts in mitosis, it is important to fully explain the pattern.

The analysis used to produce the data shown in Fig. 2c reporting the normalized FRET ratio values is not clearly described in the methods section. Please indicate the exact equation used to determine the FRET ratio%. It would also be helpful to see the non-normalized data as supplemental information. Interestingly, the internal data group from the interphase includes a number of data points that deviate significantly from the mean at least more so than the C-TERM group. Is there an explanation for this?

The display values (contrasts min and max) used to show the representative data in Fig2D needs to be included. Please clearly state that the images are displayed using identical contrasts or show the display values. The emphasis of this paper is quantitative microscopy measurements and the data needs to be quantitative.

It is reassuring to see that the Tsmod FRET efficiency is very similar to that of the new construct created here.

Please comment on why the C-TERM FRET efficiency measurements show reduced FRET for the metaphase compared with the interphase. This decrease needs to be used to correct the decrease observed for the internal tension sensor experiments.

MT drug inhibitors may provide additional support to the FRET measurements (in Figure 2 using the tsmod system) since these can be performed on the same cell before and after MT inhibition - where the force generating machinery is abolished. The colchicine control needs to be performed for the tsmod to better verify the obtained results.

Talin sensor:

As this is a novel protein sensor that is being engineered into CENP-C, the authors should confirm that CENP-C is still functional and thus that its structure is not significantly altered with its incorporation. Conversely, how accessible is the stretched talin domain when it is placed inside CENP-C protein? Will vinculin head domains have the same access for binding with the same kinetics or is binding partially impacted by its incorporation into a larger protein? In other words, how does one fully confirm that the current probe recapitulates the force values described in the original *in vitro* calibration? I would like to emphasize that the previous work by del Rio et al. was performed *in vitro* and not within a living cell. Conditions, buffers, and crowding are all different -which likely contributes to the measured forces

The RFP-GFP quantification studies require further justification or controls depending on the size of the folded CENP-C domain and where the vinculin head domain binds. GFP and RFP are a FRET pair. This is likely not a problem, however, the authors should comment that the domain is large enough that the labeled VH is binding outside the Forster radius. Otherwise, the use of the GFP:RFP ratio for quantification of VH head binding could be contaminated by unintended FRET, and depending on where the VH head binds it may have a different contribution. If the proteins are in close proximity, their FRET could be easily tested with acceptor photobleaching or by monitoring acceptor fluorescence while exciting the donor.

The characterization of samples "colchicine" or "metaphase" is not reflective of the conditions. Samples in plots should be labeled as + colchicine or - colchicine, as the + colchicine cells were treated while in metaphase.

Finally, the authors start to raise an interesting discussion about how mechanics are transmitted through this protein complex. Although it is beyond the scope of this paper, the papers significance and impact would be improved if the study addressed a biological question beyond quantification of the force. However, even as presented, the data in this paper informs our understanding of how these proteins under low force could cooperatively transmit such large forces to the kinetochore. The findings would also be useful in informing and parameterizing mathematical models for further study of metaphase molecular mechanics.

Figure 3E needs error bars.

Rebuttal

Reviewer #1

Comment 1: It seems crucial to provide at least some information about whether the sensors affected kinetochore function.

Response 1: This is a very important point (also raised by reviewer 2) and we have now added new data to figures 2 and 3 showing that chromosome alignment was completely unaffected when endogenous CENP-C was partially depleted from both the TSMOD- and TR-expressing cells. Also, we were able to build stable cell lines with every reporter construct described here and the cells grow at the same rate as WT cells. Given comment 5 below, this point was not made clearly in the original text and so in addition, to the new endogenous CENP-C depletion data, there is a more explicit statement regarding the use of stable cell lines and the absence of dominant negative effects on page 5, lines 15-17.

Comment 2: Almost no information is given about the structure of Cenp-C. What is known about its domain structure?

Response 2: We have added a disorder plot to figure 1 and provided more detail about CENP-C structure in the main text on page 5, lines 10-14.

Comment 3: Can you show brightfield images of the S2 cells? Also, it would be helpful to understand the arrangement of spindle microtubules in the metaphase cells, and why the dots are clustered even in the interphase cells (i.e., in Figures 2B and 2D).

Response 3: Unfortunately, brightfield images of these cells are really quite terrible because they are ugly little buggers. However, this is a very legitimate point (this really is not a helpful set of images!!) and so we have added new interphase images and provided some labeled landmarks.

Comment 4: The use of space in the figures seems imbalanced.

Response 4: We have followed the advice provided by the reviewer to better organize figure 2.

Comment 5: The engineered Cenp-C-sensors were expressed (using a Cenp-C promoter) from DNA constructs that were transiently transfected into the cells. Presumably the endogenous, untagged Cenp-C was also present. Could the load be unequally shared between the tagged and untagged Cenp-C molecules? This possibility is apparently not considered.

Response 5: Another very good point. In fact, we've always felt that our estimate (on the low-end) is likely an underestimate of the true amount of force and this is one of the reasons. We have added text addressing this point on page 10, lines 2-11.

Comment 6: The methods state simply that "mitotic cells were identified". On what basis?

Response 6: We look for paired sister centromeres and an absence of nuclei. This is more clearly stated in the Materials and Methods on page 15, lines 19-20.

Comment 7: To help readers less familiar with the temporal patterns of centromere localization, it would be good to be more explicit about what the normal pattern is for Cenp-C.

Response 7: This is now described in the text on page 5, lines 9-10, and page 6 lines 10-12.

Comment 8: From page 8 - "in cases where there were only 2 - 4 (rather than 12 - 13) CenpC linkages to the fluxing MT." This sentence is terribly confusing. In what situation would you expect only 2 - 4 linkages?...This speculative discussion point seems very unclear and possibly unnecessary. Is there a reference missing here?

Response 8: Well put, and agreed that this sentence was both confusing AND speculative. It has been changed to a much simpler point with appropriate referencing to read: "It is also noteworthy that 10 nM taxol is sufficient to reduce kt-MT flux by >90% in S2 cells⁴¹ and so the data do not rule out that that k-fiber flux may also contribute to force generation."

Comment 9: "...the observation that Dhc RNAi did not reduce kinetochore tension supports the conclusion that depolymerizing kt-MT plus-ends are the dominant poleward-directed force producers at bioriented Drosophila kinetochores." This seems overstated. The data are consistent with a depolymerization-based mechanism for force production, but other possibilities are not ruled out.

Response 9: See response 8 for one possibility that is now discussed more explicitly. Importantly, we have added new taxol data presented in Figure 2H that we feel more directly addresses this point. We found that addition of 500 nM taxol to dynein-depleted cells (which we need to deplete to reduce the prevalence of monopolar spindles assembled at this concentration) exhibited an increase in FRET efficiency (lower force) compared to DMSO-treated controls. It did not drop to interphase levels, which is discussed. In light of this result we have also added a discussion on the possibility that kinesin-13 motors could contribute to force generation by promoting MT catastrophes at the kinetochore.

Comment 10: "Given that k-fibers both flux and pull and that straight (presumably polymerizing) and curled (presumably depolymerizing) MTs are present..." It seems you are referring here to curled versus straight protofilaments at the ends of kinetochore-attached microtubules, not microtubules per se. This sentence needs work.

Response 10: This sentence has been simplified and now contains a reference to a paper that came out in NCB around the time of submission (perhaps even after) that begins to address the very point that I was attempting to make in this sentence the first time around.

Comment 11: In general, the final paragraph of the discussion is wordy and not very clear. Given the very clear writing in the paper leading up to this final paragraph, the latter stands out.

Response 11: The final paragraph has been restructured and reworded. I feel the evolutionary context of our force measurement is an important point to make although it must be clearly stated for it to be appreciated by the reader. If the reviewer feels the final paragraph is still unclear then we would like an opportunity to further improve it.

Comment 12: Please explain more clearly what regions were chosen for fluorescence quantification. "...using a region-in-region background subtraction...."

Response 12: A clearer explanation has been provided in the Materials and Methods on page 15, lines 24-25.

Comment 13: In Figure 3, structurally detailed views of the talin rod domain under low and high forces are shown. Where do these structures come from? Please provide a reference, or a PDB number, so that readers can understand the basis of these diagrams. The structural changes with force seem very subtle. Is there no unfolding of TR?

Response 13: This diagram is based on a theoretical structure described in a Talin review, which is now cited in the figure legend. The structural changes that occur to promote VH binding are not understood and may be quite subtle – much more than what is shown in the original del Rio et al. paper that has a diagram that is much more speculative.

Comment 14: It would be useful to put the histograms in Supplementary Figure 1 all on the same axes. Maybe merge the histograms as in Supplementary Figure 3 so the two that are supposed to be compared are on the same graph.

Response 14: The axes are now better aligned and pairwise comparisons with overlaid histograms are shown in Supplementary Figure 1.

Comment 15: In Figure 3E, the authors' data from only the test condition (with TR sensor inserted into the middle of Cenp-C) are compared to the del Rio 2009 measurements. Why not also indicate the control conditions (with C-terminal TR sensor, and with colchicine) on the same plot?

Response 15: Good suggestion - the control condition is now shown in the plot.

Reviewer #2

General comments:

Comment 1. Note that the estimated FRET efficiencies produce an ensemble average FRET efficiency, which likely includes molecules that are at a resting state and molecules that are in a tense state. Therefore, when trying to estimate the amount of applied force it is imperative to emphasize that these are somewhat lower bound estimates of tension rather than average values of tension. For example, it is possible that 10% or 50% of the FRET probes imaged are not

experiencing any force, while the remaining probes experience 6 pN of force. Therefore the average would be 2 pN. In effect, one is always estimating a minimum force that each molecule would experience to produce the observed FRET efficiency. This point needs to be clearly indicated.

Response 1: This is a very good point and we have added this language to the text on page 10, lines 2-11.

Comment 2. The original Tsmod (Nature 2010) used mTFP1 as the donor fluorescent protein. In this work, the mTurquoise2 donor was used instead. Unfortunately, this fluorophore has different optical properties (Quantum yield, extinction coefficient, lambda max, donor lifetime) and hence the J-integral and the forster distance are different from that of the fluorophores used for the Tsmod. Therefore, the Tsmod calibration can not be used directly to determine the amount of applied force. FLIM was used in the original Tsmod work to determine the QE and the extension of the sensor.

Response 2: The spider-silk reporter was calibrated using optical trapping of Cy3 and Cy5 labels flanking the flagelliform sensor (so-called TSMoDCy). In the description of the calibration there is no mention of FLIM measurements being done in the vitro calibration assay, rather the fluorescence intensities of Cy3 and Cy5 were recorded as force was applied. Live-cell TSMoDCy FRET efficiencies were determined by FLIM in cells and spectroscopy in vitro. Zero-force FRET efficiency of the two reporters and the Forster distances (along with the Forster equation) are the two key features of the reporters (TSMoDCy and TSMoDCy) that were used to translate the direct in vitro calibration of the TSMoDCy into the theoretical calibration of the TSMoDCy for live-cell force estimates. An important point made in the description of the theoretical calibration of the TSMoDCy was that "both reporters have Forster distances of ~6.0 nm." Since 1) the zero-force measurement of our mTurquoise2-containing reporter has nearly an identical FRET efficiency (~23.5%) as the original TSMoDCy, and 2) the Forster Radii of mTurquoise2/mVenus is 5.8 nm we feel that it is reasonable to use the original theoretical calibration of the mTFP1/mVenus TSMoDCy reporter to make our estimate here. A statement to this effect has been added to the materials and methods.

Comment 3. Statements that a "statistically significant 7% decrease in FRET emission ratio was measured in metaphase..." need to be backed up with a statement of the p-value and also the standard deviation or the SEM of the mean difference between the control group and the strained group.

Response 3: Statements of p-values and SEM are in the figure legends; however, we have now added statements of p-value and report SEM in the main text.

Additions/Changes made in response to FRET sensor comments: The use of the TagRFP tag was intentional and indispensable. This is because S2 cells are not amenable to typical

transmitted light imaging like DIC or phase as they are ugly and mitotic cells cannot be readily identified. Thus, if we only had the TSMoD available to us then we would have to scan our samples on either the CFP or YFP channels. This would present a significant problem as we are confident from imaging numerous CFP variant/YFP variant reporters that photo-bleaching of the fluorophores is a major problem in collecting FRET data. In fact, we see dramatic and artifactual (based on controls we've done) changes in the FRET ratio even after relatively short exposure times during time-lapse imaging. (Side note: I am now convinced that FLIM is the only way to go for time-lapse imaging of CFP/YFP FRET reporters.) All of this is to say that we designed the reporter with the TagRFP-T tag to achieve the absolute minimum exposure (one time and move on) of the TSMoD. While we confirmed that there was minimal bleed-through from the RFP before doing the FRET or acceptor photo-bleaching experiments, the point raised here is a very good one that made us go back and carefully quantify the bleed-through from the TagRFP. **The new data has been added to Supplemental Figures 1 and 3** and show that there is ~0 – 0.5% bleed-through from RFP into the CFP, YFP and FRET channels for the wide-field FRET imaging conditions. Furthermore, there was 0 detectable bleed-through from RFP into the CFP channel, which is the only channel that is used to calculate the FRET efficiency, for the acceptor photobleaching imaging conditions. The RFP bleed-through into the YFP channel was also negligible at ~0.6% although this channel is only imaged to confirm the efficiency of the acceptor photobleaching. Regarding, figure 2B (now 2C) this was not meant to be quantitative but rather qualitative in nature to show the readers that the localization pattern of CENP-C is normal and all the fluorophores are visible. We have added new interphase images and provided some labeled landmarks. All scaling is matched between the appropriate channels and this is indicated in the legends with regards to both 2B (now 2C) and 2D (now 2E). Regarding the application of a correction to the internal TSMoD data based on the C-TERM acceptor photobleaching measurements, I am uncomfortable doing this because the C-TERM interphase and metaphase data are not statistically significantly different ($p > 0.05$) from each other nor from the internal TSMoD interphase measurements. In the text we state that the FRET ratio data is an indicator of more force being applied to the kinetochore during metaphase. In general, the FRET ratio measurements are presented as (and really were) the initial indicator of there being a measurable amount of force being applied to the sensor. We never felt the data was sufficient to reach major conclusions about the magnitude of force; therefore, these data serve as an experimental segue into what we consider to be the more relevant measurements of FRET efficiencies (in terms of estimating the force applied) from the acceptor photo-bleaching experiments. Regarding the colchicine measurements, we have observed that colchicine exhibits some fluorescence in the CFP channel when bound to tubulin and we are also uncomfortable with FRET-based imaging cells before and after drug treatment for the reasons stated above regarding concerns about photobleaching. Furthermore, it would not be technically feasible to collect enough data with drug wash-in conditions because it is incredibly low throughput.

Additions/Changes made in response to talin sensor comments:

We have added new data to figures 2 and 3 showing that both TSMoD- and TR-expressing cells depleted of 50-60% of endogenous CENP-C align their chromosomes equally as well as control RNAi-treated cells. Also, we were able to build stable cell lines with every reporter

construct described here and there were no dominant negative effects. This is now stated more explicitly page 5, lines 15-17. Regarding the accessibility of the TR domain and the comparison of in vitro and live cell measurements, the TR is inserted into the central unstructured region of CENP-C (now discussed on page 5, lines 9-15) and we also now clearly state and discuss on page 9, lines 12-15 that the del Rio et al. measurements were done in vitro as opposed to our live-cell measurements. We also added the point that we feel the comparison is validated by the fact that our measurement of VHs bound in cells in the low/no force conditions are comparable to the zero-force measurements made in vitro in del Rio et al (now shown in Figure 3F at suggestion of reviewer 1). Regarding the possibility of FRET between RFP and GFP, **we have added new data to supplemental figure 3** showing that there is no detectable FRET between the VH-EGFP and TagRFP tag on CENP-C in either metaphase or colchicine-treated cells. The bleed-through from GFP and RFP into the FRET channel was determined for the imaging conditions. We then monitored RFP fluorescence while exciting GFP in metaphase and colchicine-treated, CENP-C internal TR reporter/VH-EGFP-expressing cells and applied bleed-through and background corrections. A description of this is now provided in the materials and methods. Regarding the characterization of samples as "colchicine" or "metaphase" we have re-labeled the colchicine as "+ colchicine" but we have not changed the metaphase designation as it is not correct that the "+ colchicine" cells were treated while in metaphase. These cells were treated for ~1.75 hours prior to imaging and so, while some cells may have previously been in metaphase, many of the cells that were imaged likely entered mitosis in the presence of colchicine. Regarding the comment that Figure 3E needs error bars, the error bars were in the original figure but they were difficult to see. They have now been made wider and changed to white with black outlines so we hope they show up better.

Reviewers' Comments:

Reviewer #1 (Remarks to the Author):

In my view the revised paper is suitable for publication in Nature Communications. In particular, the authors have addressed the concern about whether insertion of the sensors into Cenp-C affected kinetochore function. (Apparently not.) There is also nice new data showing how a low dose of taxol affects sensor output. Many changes to the text and figures have been made to improve readability and accessibility to a wide audience. I congratulate the authors on a fine manuscript.

Reviewer #2 (Remarks to the Author):

The revision is sufficient and the response was thorough. Based on updated discussion and new data, I would recommend the paper for publication. The added comment about the tension free cases having the same number of binding vinculin heads and the in vitro study makes that section more compelling. The discussion also better addresses the issue of underestimating the applied force due to ensemble averaging, etc. The revised discussion is improved. It adds significance and impact to the paper that was missing/poorly explained in the original submission. The figures are improved but still require further improvements in organization and display for increased clarity. It is unfortunate that brightfield images can not be included, but it is at least helpful to see the outline of the cell, which is now included. The space still seems not fully used in the figures but this can be further improved.